# Circulating microRNA Profiles for Premature Cardiovascular Death in Patients with Kidney Failure with Replacement Therapy

**DOI:** 10.3390/jcm12155010

**Published:** 2023-07-30

**Authors:** Canan Kuscu, Yamini Mallisetty, Surabhi Naik, Zhongji Han, Caleb J. Berta, Cem Kuscu, Csaba P. Kovesdy, Keiichi Sumida

**Affiliations:** 1Transplant Research Institute, Department of Surgery, University of Tennessee Health Science Center, Memphis, TN 38163, USA; snaik2@uthsc.edu (S.N.); ckuscu1@uthsc.edu (C.K.); 2Division of Nephrology, Department of Medicine, University of Tennessee Health Science Center, Memphis, TN 38163, USA; ymallise@uthsc.edu (Y.M.); zhan10@uthsc.edu (Z.H.); cberta@uthsc.edu (C.J.B.); ckovesdy@uthsc.edu (C.P.K.); 3Nephrology Section, Memphis VA Medical Center, Memphis, TN 38104, USA

**Keywords:** cardiovascular disease, circulating small non-coding RNAs, chronic kidney disease, end-stage kidney disease, miRNAs, mortality, RNA-seq

## Abstract

Introduction: Patients with kidney failure with replacement therapy (KFRT) suffer from a disproportionately high cardiovascular disease burden. Circulating small non-coding RNAs (c-sncRNAs) have emerged as novel epigenetic regulators and are suggested as novel biomarkers and therapeutic targets for cardiovascular disease; however, little is known about the associations of c-sncRNAs with premature cardiovascular death in KFRT. Methods: In a pilot case-control study of 50 hemodialysis patients who died of cardiovascular events as cases, and 50 matched hemodialysis controls who remained alive during a median follow-up of 2.0 years, we performed c-sncRNAs profiles using next-generation sequencing to identify differentially expressed circulating microRNAs (c-miRNAs) between the plasma of cases and that of controls. mRNA target prediction and pathway enrichment analysis were performed to examine the functional relevance of differentially expressed c-miRNAs to cardiovascular pathophysiology. The association of differentially expressed c-miRNAs with cardiovascular mortality was examined using multivariable conditional logistic regression. Results: The patient characteristics were similar between cases and controls, with a mean age of 63 years, 48% male, and 54% African American in both groups. We detected a total of 613 miRNAs in the plasma, among which five miRNAs (i.e., miR-129-1-5p, miR-500b-3p, miR-125b-1-3p, miR-3648-2-5p, and miR-3150b-3p) were identified to be differentially expressed between cases and controls with cut-offs of *p* < 0.05 and log2 fold-change (log2FC) > 1. When using more stringent cut-offs of *p*-adjusted < 0.05 and log2FC > 1, only miR-129-1-5p remained significantly differentially expressed, with higher levels of miR-129-1-5p in the cases than in the controls. The pathway enrichment analysis using predicted miR-129-1-5p mRNA targets demonstrated enrichment in adrenergic signaling in cardiomyocytes, arrhythmogenic right ventricular cardiomyopathy, and oxytocin signaling pathways. In parallel, the circulating miR-129-1-5p levels were significantly associated with the risk of cardiovascular death (adjusted OR [95% CI], 1.68 [1.01–2.81] for one increase in log-transformed miR-129-1-5p counts), independent of potential confounders. Conclusions: Circulating miR-129-1-5p may serve as a novel biomarker for premature cardiovascular death in KFRT.

## 1. Introduction

Kidney failure with replacement therapy (KFRT) is a condition characterized by an extremely high risk of cardiovascular morbidity and mortality, with almost half of all deaths attributable to cardiovascular disease [1]. Despite considerable efforts to improve cardiovascular outcomes, the substantial cardiovascular disease burden in KFRT remains unresolved, consuming a disproportionate amount of financial resources [2,3]. Therefore, identifying novel risk factors and promising biomarkers for cardiovascular disease in KFRT is of critical importance toward the development of novel preventives and therapeutic approaches to premature cardiovascular death in this population.

Small non-coding RNAs (sncRNAs) are a group of small (18–200 nucleotide-long) RNA molecules that function as epigenetic regulators of gene expression at the post-transcriptional level involved in intercellular communication and crosstalk between different organs [4]. As key regulators of homeostasis, their dysregulation underlies several morbidities through the combinatorial effect of gene expression changes in all related downstream targets [5,6,7,8,9]. MicroRNAs (miRNAs), for example, are the most extensively studied class of sncRNAs that negatively regulate gene expression by partially pairing to the 3′-untranslated region of their target messenger RNAs, leading to translation repression and/or transcript degradation [10]. The other major classes of sncRNAs include transfer RNA (tRNA), small interfering RNAs (siRNAs), and piwi-interacting RNAs (piRNAs) [11,12,13].

While the majority of sncRNAs exist intracellularly, they can also be found in various body fluids, including blood, through passive leakage and/or active secretion from cells [14,15,16,17,18]. Importantly, accumulating evidence indicates that the aberrant expression of these so-called circulating sncRNAs (c-sncRNAs) is associated with the risk of cardiovascular disease [19,20,21,22,23,24,25], suggesting the potential of c-sncRNAs as novel cardiovascular biomarkers and therapeutic targets [26,27]. However, the existing studies on the association of c-sncRNA with cardiovascular disease are predominantly from the non-KFRT populations, limiting the evidence among patients with KFRT who display a distinct cardiovascular phenotype [28]. Furthermore, a few studies reporting the possible involvement of circulating miRNAs (c-miRNA) in cardiovascular health in patients with KFRT have focused primarily on the association of pre-specified c-miRNAs (e.g., miR-125b and miR-133a) with cardiovascular surrogates, such as vascular calcification and cardiac hypertrophy [29,30,31]; hence, it remains unclear if these c-miRNAs are associated with hard cardiovascular outcomes, including premature cardiovascular death, in KFRT.

We hypothesized that c-miRNAs would be differentially expressed between patients with KFRT who died of a cardiovascular event and those without such an event and that the differentially expressed c-miRNAs would be involved in cardiovascular pathophysiology that could lead to premature cardiovascular death in patients with KFRT. In this pilot case-control study of patients with KFRT receiving hemodialysis therapy, we therefore aimed to perform comprehensive profiling of c-miRNAs and identify differentially expressed c-miRNAs associated with premature cardiovascular death in these patients.

## 2. Methods and Materials

### 2.1. Study Design

This was a case-control study sourced from a prospective study of anonymized samples and statistically de-identified clinical data (detailed below) obtained from a biorepository assembled by DaVita Clinical Research (Minneapolis, MN, USA). Anonymized samples and statistically de-identified data were made available to the authors for academic research via a grant program called BioReG.

### 2.2. Study Population

The DaVita Clinical Research biorepository comprises blood samples and clinical data from 4028 individuals with prevalent end-stage renal disease who received hemodialysis at a large dialysis organization (LDO) between May 2011 and October 2013, as previously described [32]. Patients with hemoglobin < 8.0 g/dL, who were <18 years of age, who were pregnant, or who had any physical, mental, or medical condition, which prohibited the ability to provide informed consent were excluded from participation. The biorepository sampling protocol was reviewed and approved by an Institutional Review Board (IRB) (Quorum IRB, Seattle, WA, USA) and patients provided written informed consent prior to the initiation of sample collection.

For the present pilot case-control study, we used plasma samples at baseline (i.e., first blood sampling date) and clinical data corresponding to the blood sampling date in a total of 100 hemodialysis patients within the repository housed at the University of Tennessee Health Science Center (UTHSC) (UT-DaVita hemodialysis cohort; *n* = 978) [33,34]. Cases (*n* = 50) were hemodialysis patients who died of a cardiovascular event, while controls (*n* = 50) were those who remained alive over the entire follow-up, matched 1:1 by age, sex, race, and dialysis vintage to account for major non-modifiable cardiovascular risk factors. The study was approved by the IRB at UTHSC (IRB protocol number: 17-05299-XP).

### 2.3. Biorepository Biospecimen and Clinical Data Collection

Under the biorepository study protocol, blood samples were collected from each subject at baseline and, thereafter, every 3 months for up to one year. Pre-dialysis blood samples were collected and processed according to a standardized protocol as previously described [32,33,34]. Briefly, anonymized plasma samples were shipped in refrigerated packs from the centralized laboratory to the researchers and stored at −80 °C. Clinical data for each biorepository subject were collected by the LDO during the course of routine care and were maintained in the LDO electronic health record. The data were then provided to the researchers by DaVita Clinical Research in a statistically de-identified form. Cardiovascular death was defined as death caused by acute myocardial infarction, atherosclerotic heart disease, cardiomyopathy, cardiac arrhythmia, cardiac arrest, or congestive heart failure [34].

### 2.4. RNA Purification from Plasma and cDNA Library Preparation

Total RNA from the plasma samples were purified as described before [35]. Briefly, c-sncRNAs were purified from 200 µL plasma using the Qiagen miRNeasy Serum/Plasma Advanced Kit (Cat No. 217204). The purified sncRNAs were then eluded with 15 µL RNase-free water, and 5 µL of RNA was used for c-sncRNA library using NEBNext Small RNA Library Prep Set (Cat No. E7330L) according to the manufacturer’s protocol. In total, 18 cycles of PCR amplification were performed. c-sncRNA libraries were purified by polyacrylamide gel and then subjected to Agilent bioanalyzer for quality control. Equal amounts of libraries were mixed and sequenced in NextSeq500 in Oklahoma Medical Research Foundation.

### 2.5. c-sncRNA Sequencing Data and Differential Expression Analysis

c-sncRNA sequencing was performed for 100 hemodialysis patients with (*n* = 50) and without (*n* = 50) a fatal cardiovascular event. Obtained reads were mapped to human genome using unitas tool [36]. We obtained around 7 million reads per sample on average and 60% of obtained reads mapped to human genome on average. Differentially expressed c-miRNAs were analyzed using DeSeq2 tool [37], and *p*-adjusted values for differentially expressed genes were calculated in DESeq2 using the Benjamini-Hochberg method. Detected c-miRNAs and differentially expressed c-miRNAs can be found in Appendix A.

### 2.6. mRNA Target Prediction and Pathway Enrichment Analysis

mRNA targets of miR-129-1-5p were predicted using TargetScan tool [38]. In total, 732 mRNA targets with conserved sites (Appendix A) were used in downstream analysis. Pathway enrichment analysis such as for Kyoto Encyclopedia of Genes and Genomes (KEGG) and Reactome Pathways was performed using EnrichR (https://maayanlab.cloud/Enrichr/ (accessed on 15 December 2022)) [39]. Phenotype Genotype Integrator (PhenGenI) in Diseases/Drugs section in Enrich R website was used to investigate the relationship between miR-129-5p and the patient cohort using targets of miR-129-5p.

### 2.7. Statistical Analysis

Baseline patient characteristics by cardiovascular case status were presented as numbers (percentages) for categorical variables and mean (standard deviation [SD]) for continuous variables with a normal distribution or median (interquartile interval [IQI]) for those with a skewed distribution. Differences between groups were assessed using Chi-square test or Fisher’s exact test (for categorical variables) and t-test or Wilcoxon rank-sum test (for continuous variables), as appropriate.

In order to examine the risk of cardiovascular death associated with differentially expressed miR-129-1-5p, we applied multivariable conditional logistic regression models. Given the limited sample size of this study, the exposure (i.e., the log-transformed miR-129-1-5p counts) was treated as a continuous variable, and the following incremental models were used to account for potential confounders on top of the matching factors (i.e., age, sex, race, and dialysis vintage) based on theoretical consideration, data availability, and residual imbalance after matching: Model 1 was unadjusted; Model 2 included body mass index, diabetes mellitus, congestive heart failure, and ischemic heart disease; and Model 3 was additionally adjusted for serum alkaline phosphatase and phosphorus levels. All analyses were performed in patients with complete data available using STATA/MP Version 17 (STATA Corporation, College Station, TX, USA). A threshold of statistical significance was set at the level of *p* < 0.05 for these association analyses.

## 3. Results

### 3.1. Baseline Characteristics

Patients’ baseline characteristics by cardiovascular case status are presented in Table 1. Cases and controls were of similar age at baseline (means of 63.1 ± 11.0 and 63.1 ± 11.2 years, respectively) and, by design, did not differ for other matching factors, including sex (48.0% male in both), self-reported race/ethnicity (54.0% African American in both), and dialysis vintage (4.1 ± 3.4 and 4.4 ± 3.2 years, respectively). Compared with controls, cases had a lower prevalence of erythropoiesis-stimulating agent use and tended to have higher serum alkaline phosphatase levels and a higher prevalence of statin use.

### 3.2. c-sncRNA Profiles in Hemodialysis Patients with and without Cardiovascular Death

Using the plasma of 50 hemodialysis patients who died of cardiovascular events as cases and that of 50 matched hemodialysis controls, we detected 613 unique c-miRNAs (Appendix A). Among these, miR-423-5p, miR-122-5p, and miR-486-5p were the most abundant c-miRNAs in these patients (Appendix A). We identified five differentially expressed c-miRNAs, including miR-129-1-5p, miR-500b-3p, miR-125b-3p, miR3648-2-5p, and miR3150b-3p, with cut-offs of *p* < 0.05 and log2 fold-change (log2FC) > 1 (Figure 1A and Appendix A). Interestingly, all these miRNAs were upregulated in the cases compared to the controls. When we applied a more stringent cut-off of *p*-adjusted < 0.05 and log2FC > 1, miR-129-1-5p was the only miRNA significantly differentially expressed between groups (Figure 1B and Appendix A).

### 3.3. Pathway Enrichment Analysis of Circulating miR-129-5p

To understand the functional role of c-miRNAs in premature cardiovascular death in patients with KFRT, we performed a pathway analysis using the predicted targets of miR-129-5p. The top three enriched KEGG pathways in miR-129-5p targets were adrenergic signaling in cardiomyoctes, arrhythmogenic right ventricular cardiomyopathy, and oxytocin signaling pathway (Figure 2). Intriguingly, all these pathways were reported to be directly related to heart function [40,41,42]. Moreover, the most enriched Reactome pathway in miR-129-5p targets was transcriptional regulation by Methyl-CpG-Binding Protein 2 (MECP2), which was shown to have a critical role in heart failure [43,44,45] (Figure 3). In parallel, atherosclerosis was the most enriched phenotype based on the Phenotype Genotype Integrator Enrichment using Enrich R (Figure 4).

### 3.4. Association of Circulating miR-129-1-5p with Cardiovascular Death

Since miRNAs were the most abundant c-sncRNA species in the sequencing results, and only miR-129-1-5p was significantly differentially expressed between cases and controls when using stringent significance thresholds (i.e., *p*-adjusted < 0.05 and log2FC > 1), only miR-129-1-5p was considered in the association analysis with cardiovascular death. Table 2 shows the association of circulating miR-129-1-5p with cardiovascular death using univariable and multivariable conditional logistic regression analyses. In the univariable model, circulating miR-129-1-5p was associated with cardiovascular death, albeit without reaching statistical significance (odds ratio [OR] [95% confidence interval (CI)] for 1-unit higher log-transformed circulating miR-129-1-5p counts: 1.28 [0.89–1.86], in Model 1). This association was slightly modified after multivariable adjustment, with a statistically significant association observed in a fully adjusted model accounting for body mass index, diabetes mellitus, congestive heart failure, ischemic heart disease, and serum alkaline phosphatase and phosphorus levels, on top of the matching factors (i.e., age, sex, race, and dialysis vintage) (adjusted OR [95% CI]: 1.68 [1.01–2.81], in Model 3; Table 2).

## 4. Discussion

In this pilot case-control study of 100 patients with KFRT receiving maintenance hemodialysis, we detected a total of 613 miRNAs in the plasma, among which five miRNAs (i.e., miR-129-1-5p, miR-500b-3p, miR-125b-1-3p, miR-3648-2-5p, and miR-3150b-3p) were differentially expressed between those who died of a cardiovascular event and those who remained alive. When using stringent statistical thresholds of *p*-adjusted < 0.05 and log2FC > 1, only miR-129-1-5p remained differentially expressed, with significantly higher expression levels seen in the cardiovascular cases than in the controls. Pathway enrichment analysis using predicted miR-129-1-5p mRNA targets demonstrated enrichment in adrenergic signaling in cardiomyocytes, arrhythmogenic right ventricular cardiomyopathy, and oxytocin signaling pathways. Furthermore, we found that higher circulating miR-129-1-5p levels were significantly associated with a higher risk of cardiovascular death, independent of potential confounders.

Over the past decade, an increasing number of c-sncRNAs have been identified as biomarker candidates in cardiovascular disease [46]; however, their biomarker potential has been poorly understood among patients with KFRT who display a unique cardiovascular phenotype [28]. In an earlier study of 64 maintenance hemodialysis patients and 18 healthy controls, researchers compared the plasma concentrations of circulating miR-133a between hemodialysis patients and healthy controls and examined their association with cardiac hypertrophy among those on hemodialysis. They demonstrated that the circulating miR-133a levels were lower in the hemodialysis patients than in the healthy controls, and they were negatively associated with left ventricular mass index and interventricular septum thickness in hemodialysis patients [30]. Similarly, the levels of circulating miR-125b and miR-206 and their association with vascular calcification among hemodialysis patients have been reported in subsequent studies [29,47]. In a recent study evaluating the association of circulating levels of a selected miRNAs panel (30a-5p, 23a-3p, 451a, and let7d-5p) with a composite of all-cause and cardiovascular mortality in 74 maintenance hemodialysis patients, lower (vs. higher) levels of circulating miRNA 30-5p, 23a-3p, and 451a were significantly associated with a higher risk of the composite outcome [48]. Importantly, however, all these previous studies assessed pre-specified miRNAs and their associations with surrogate measures or composite outcomes of cardiovascular disease, limiting the ability to identify previously unknown or under-recognized circulating miRNAs that may be involved in the pathogenesis of premature cardiovascular death in KFRT. In this context, our unbiased (hypothesis-free), comprehensive profiles of c-miRNAs using a matched case-control design would be of particular value, allowing the identification of novel c-miRNAs associated with premature cardiovascular death in patients with KFRT. And in fact, among 613 c-miRNAs detected in the plasma of 50 cardiovascular cases and 50 controls in our study, we found that circulating miR-129-1-5p was the only c-miRNA that was significantly differentially expressed between groups (Figure 1) and was also independently associated with premature cardiovascular death in these patients (Table 2).

MiR-129-1-5p is a 5′-prime product of a member of the miR-129 family, miR-129-1, which is located on chromosome 7q32.1 and has been shown to be involved in various oncological and non-oncological diseases, including cardiovascular disease [49]. In previous experimental studies in heart failure, for example, miRNA-129-5p (same as miRNA-129-1-5p) [50] has been shown to play a protective role in myocardial cell injury through targeting high-mobility group box-1 (HMGB1) and tumor necrosis factor receptor-associated factor 3 (TRAF3) and thereby ameliorating oxidative stress and inflammatory responses in cardiomyocytes [51,52]. Although these observations seemingly contradict what we have found in the present study, the significantly higher expression of circulating miR-129-1-5p among patients with (vs. without) a fatal cardiovascular event in our study may reflect the physiological process of upregulating miR-129-1-5p in response to damaged cardiomyocytes that eventually led to their fatal cardiovascular events. Nonetheless, it is important to note that the upregulation of miR-129-5p has also been implicated in the development of atherosclerosis by impairing the protective effects of endothelial cell autophagy through miR-129-5p-mediated Beclin-1 suppression [53]. In line with this observation, our enrichment analysis using the Phenotype Genotype Integrator demonstrated that atherosclerosis was the phenotype most significantly associated with miR-129-5p (Figure 4). In addition, the top three enriched pathways in KEGG pathway enrichment suggested that miR-129-5p is directly related to cardiomyocytes and cardiomyopathy (Figure 2). Moreover, Reactome pathways enrichment analysis demonstrated that MECP2-dependent transcriptional regulation might play an important role in the pathogenesis of premature cardiovascular death in patients with KFRT (Figure 3). Interestingly, it has been shown that Mecp2 mutant mice developed lethal cardiac arrhythmias [43]. Most importantly, these findings provide novel insights into the mechanisms underlying premature cardiovascular death in KFRT and could lead to the discovery of novel molecular-based biomarkers and potential therapeutic strategies for premature cardiovascular death in this relevant population.

Despite the strengths of our study, including a well-designed matched case-control cohort nested from a nationwide prospective hemodialysis cohort, the study results must be interpreted in light of several limitations. Given the substantial heterogeneity of hemodialysis patients with various etiologies and comorbidities, our study population was not representative of all patients with KFRT. Although we adjusted for various potential confounders in the analysis to examine the association of miR-129-1-5p with cardiovascular death, due to the small sample size of this pilot case-control study, we were unable to fully account for potential confounders and hence cannot eliminate the possibility of unmeasured confounders that might have affected the association. Similarly, due to the small sample size of this study, we were unable to examine specific causes of CV death. Despite efforts to perform pathway enrichment analyses, we cannot infer any causal relationship between miR-129-1-5p and cardiovascular death, which needs to be examined in more in-depth experimental and clinical studies.

In conclusion, in this pilot case-control study of prevalent hemodialysis patients, we detected a total of 613 miRNAs in the plasma and found that circulating miR-129-1-5p was significantly upregulated (*p*-adjusted < 0.05 and log2FC > 1) among those with (vs. without) a fatal cardiovascular event. We also demonstrated in the pathway enrichment analysis that miR-129-1-5p is closely involved in cardiovascular pathophysiology. Although these findings should be validated in future larger studies, our results suggest a biomarker potential of miR-129-1-5p for premature cardiovascular death in patients with KFRT.

## Figures and Tables

**Figure 1 jcm-12-05010-f001:**
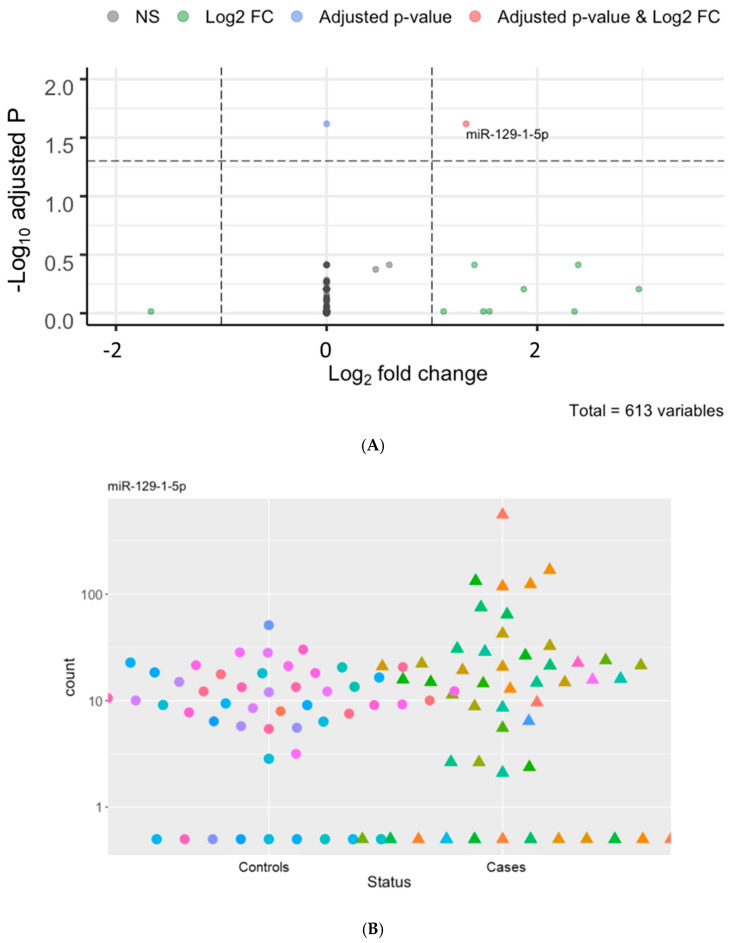
miR-129-1-5p was overexpressed in the plasma of hemodialysis patients with a fatal cardiovascular death (cases) compared to those without such an event (controls). (**A**): Volcano plot showing differentially expressed miRNAs. (**B**): Read counts for miR-129-1-5p in every patient grouped by cases (triangles) vs. controls (circles). Abbreviations: log2 FC = log2 fold-change; miRNA = microRNA.

**Figure 2 jcm-12-05010-f002:**
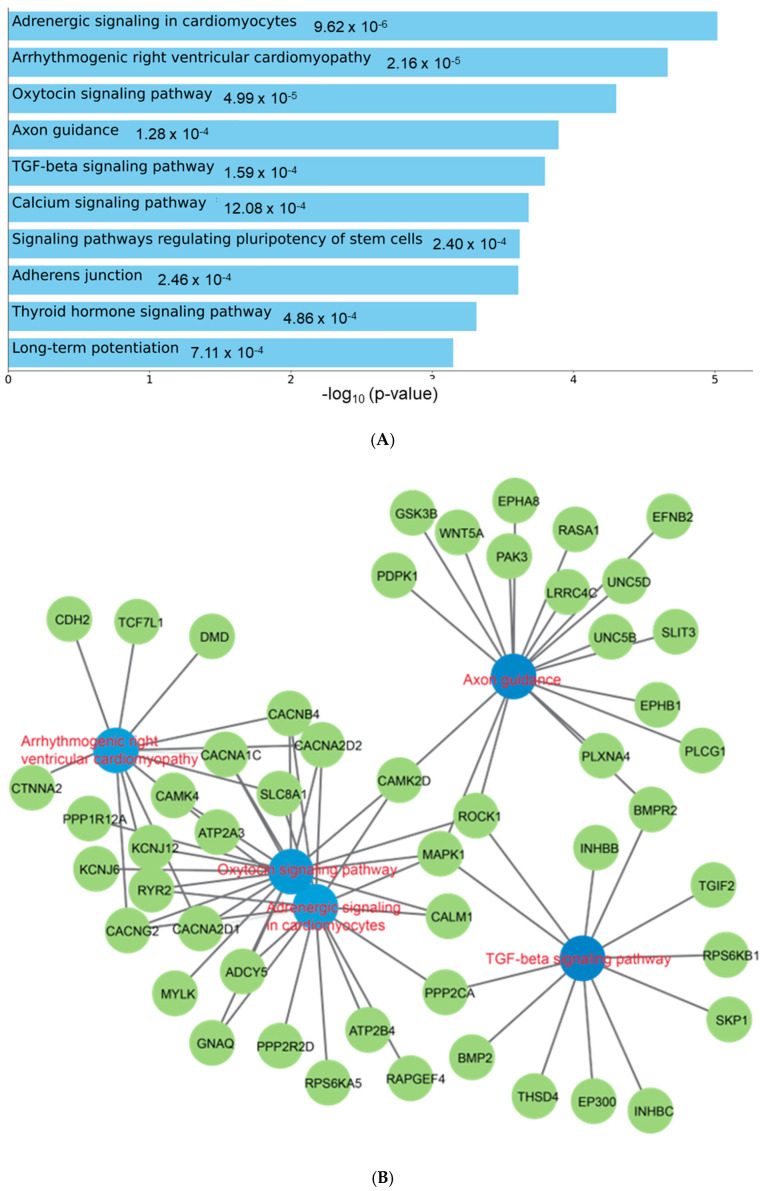
Pathway enrichment analysis for miR-129-5p targets. Enriched KEGG pathways indicated a potential relation to cardiovascular disease through cardiomyoctes. (**A**): Bar chart showing the enriched KEGG pathways. (**B**): Network depicting the miR-129-5p targets in the enriched pathways.

**Figure 3 jcm-12-05010-f003:**
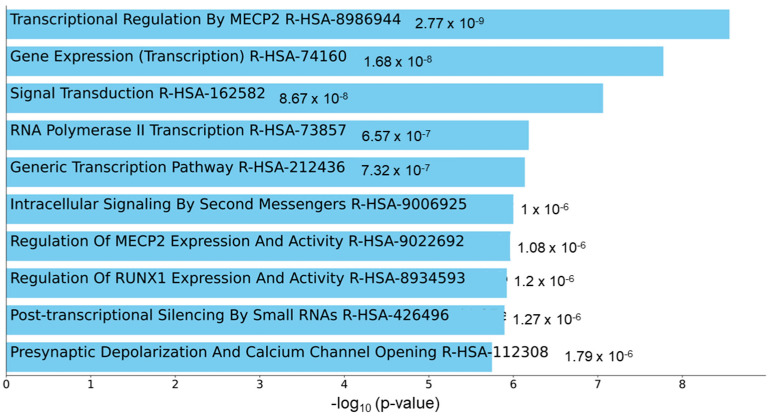
Enriched Reactome pathways hinted contribution of MECP2 dependent transcriptional regulation to fatal cardiovascular events. Bar chart highlighting the enriched pathways in Reactome pathway enrichment analysis with miR-129-5p targets. Abbreviation: MECP2 = methyl-CpG binding protein 2.

**Figure 4 jcm-12-05010-f004:**
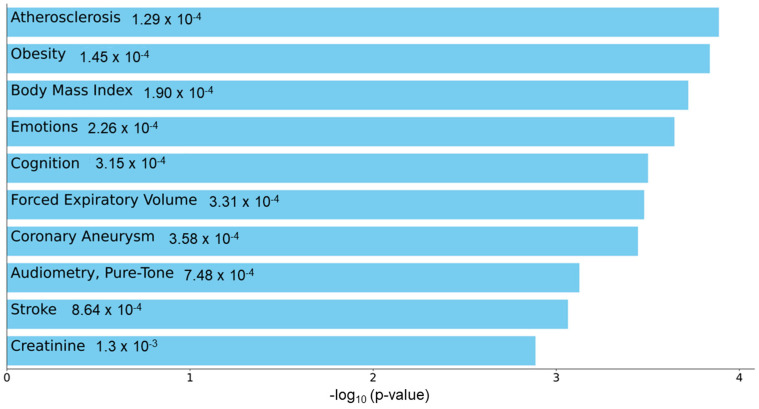
Enrichment analysis for miR-129-5p targets using Enrich R Disease/Ontologies tools. Enrichment in Phenotype-Genotype Integrator (PhenGenI), which merges genome-wide association studies (GWAS) with several other catalog databases including Gene, the database of Genotypes and Phenotypes (dbGAP), Online Mendelian Inheritance in Man (OMIM), expression quantitative trait loci (eQTL) and Single Nucleotide Polymorphism Database (dbSNP) showed potential relationship to atherosclerosis and obesity. Bar chart highlighting the enriched phenotypes in Phenotype Genotype Integrator enrichment analysis with miR-129-5p targets.

**Table 1 jcm-12-05010-t001:** Baseline patient characteristics by cardiovascular case status.

Characteristics	Cases (*n* = 50)	Controls ^a^ (*n* = 50)	*p*
Age (years)	63.1 ± 11.0	63.1 ± 11.2	1.00
Male sex	24 (48.0)	24 (48.0)	1.00
Race			1.00
White	17 (34.0)	17 (34.0)	
African American	27 (54.0)	27 (54.0)	
Others	6 (12)	6 (12)	
Dialysis vintage (years)	4.1 ± 3.4	4.4 ± 3.2	0.70
Vascular access type			0.67
Arteriovenous fistula	30 (60.0)	32 (64.0)	
Arteriovenous graft	11 (22.0)	8 (16.0)	
Catheter	9 (18.0)	9 (18.0)	
Body mass index (kg/m^2^)	30.4 ± 8.4	31.2 ± 6.4	0.60
Systolic BP (mmHg)	151.9 ± 26.9	149.4 ± 25.8	0.62
Diastolic BP (mmHg)	76.8 ± 14.7	77.2 ± 15.7	0.90
Charlson Comorbidity Index	5.8 ± 1.7	5.6 ± 1.9	0.58
Comorbidities			
Diabetes mellitus	31 (62.0)	31 (62.0)	1.00
Ischemic heart disease	6 (12.0)	5 (10.0)	0.75
Congestive heart failure	12 (24.0)	5 (10.0)	0.062
Liver disease	1 (2.0)	1 (2.0)	1.00
HIV/AIDS	0 (0)	0 (0)	1.00
Malignancies	0 (0)	0 (0)	1.00
Laboratory parameters			
Blood hemoglobin (g/dL)	11.1 ± 1.4	10.8 ± 1.0	0.13
Serum albumin (g/dL)	3.9 ± 0.4	3.9 ± 0.3	0.91
Serum calcium (mg/dL)	9.3 ± 0.7	9.1 ± 0.7	0.29
Serum phosphorus (mg/dL)	5.6 ± 1.6	5.1 ± 1.2	0.09
Serum ALP (U/L)	117.8 ± 70.7	94.8 ± 39.3	0.05
Serum intact PTH (pg/mL)	374 [225, 612]	301 [220, 504]	0.10
Medications			
Statins	18 (36.0)	10 (20.0)	0.075
ESAs	36 (72.0)	44 (88.0)	0.045
Phosphate binders	29 (58.0)	30 (60.0)	0.83
Vitamin D analogs	44 (88.0)	47 (94.0)	0.30
Aspirin	14 (28.0)	5 (10.0)	0.022
Opioids	17 (34.0)	14 (28.0)	0.52

Note: Data are presented as number (percentage), mean ± SD, or median [interquartile interval]. ^a^ Matched by age, sex, race, and dialysis vintage. Abbreviations: ALP = alkaline phosphatase; BP = blood pressure; ESAs = erythropoiesis-stimulating agents; HIV/AIDS = human immunodeficiency virus/acquired immunodeficiency syndrome; PTH = parathyroid hormone.

**Table 2 jcm-12-05010-t002:** Odds ratios and 95% confidence interval for cardiovascular death associated with circulating miR-129-1-5p expression in 100 hemodialysis patients.

Models	Odds Ratio (95% CI) *	*p*
Model 1	1.29 (0.89–1.86)	0.18
Model 2	1.44 (0.90–2.32)	0.13
Model 3	1.68 (1.01–2.81)	0.048

Note: All models matched for age, sex, race, and dialysis vintage. Model 1 is unadjusted: Model 2 is adjusted for body mass index, diabetes mellitus, congestive heart failure, and ischemic heart disease; and Model 3 is adjusted for the variables in Model 2 plus serum alkaline phosphatase and phosphorus levels. * per 1 unit increase in log-transformed circulating miR-129-1-5p counts. Abbreviations: CI = confidence interval.

## Data Availability

The data used in these analyses were provided by DaVita Clinical Research. Requests for access to data can be made in writing to DaVita Clinical Research.

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
