# Peer review of "Circulating microRNA Profiles for Premature Cardiovascular Death in Patients with Kidney Failure with Replacement Therapy"

_jcm, 2023, doi:10.3390/jcm12155010_

Round 1

Reviewer 1 Report

Overall, this is a paper which is relevant to the special issue, and which should be of interest to physicians in general.

A few points:

1. I could not find any evidence of figures (apart from legends) in either the manuscript that was supplied for review or the supplementary material file (which contained only the supplementary tables). Please supply these.

2. Abstract (line 40) - based on the KDIGO nomenclature for kidney function and disease (Levey AS et al, Kidney Int 2020; 97:1117-29), the term "kidney failure" is preferred over "end stage kidney disease" and should be used instead.

3. How did the authors calculate p-adjusted - I presumed Bonferroni correction, but this needs to be specified.

4. The sentence starting "Since MiRNAs were the most abundant c-sncRNA species........" (line 181) seems out of place insofar as it is describing results within a methods section. This should be "re-homed" within the relevant section of the results.

5. Why was there adjustment for serum alkaline phosphatase in the logistic regression models but no adjustment for haemoglobin?

6. In their pathway enrichment analysis, the authors state that the top three enriched pathways directly related to heart function. miR-129-5p has been implicated in a diverse array of human disease ranging from various cancers to MND, so were the authors referring to all reactome pathways in general or the subset likely to be related to CV disease.

7. The authors show that the association of miR-129-1-5p with CV death only became statistically significant in model 3 (table 2). This model incorporated Alk phos and phosphate, implying that one of these variables had been a confounder in the preceding models. Which one (ALP or phosphate) was the relevant variable and can the authors

Reviewer 2 Report

Comments to manuscript JCM-2499365

This is a study in which c-miRNAs profile in patients with ESKD with hemodialysis therapy was analyzed. Writing is clear and clean. A specific miRNA showed to be potentially associated with early cardiovascular death in the analyzed patients, although p-value was indeed in borderline. Nevertheless, the authors discussed well this limitation and set basis for more studies. Manuscript could be accepted. This reviewer has only a question:

1.     What was the distribution of the circulating miR-129-1-5p according to specific death cause? 

2.    I was not able to see any figure.

Reviewer 3 Report

The introduction does not list the topics of the title: Circulating microRNA AND cardiovascular death AND stage kidney disease..

.

Reviewer 4 Report

In their manuscript, Kuscu et al. evaluated circulating micro RNAs (c-miRNAs) in plasma of dialysis patients with or without cardiovascular death (cardiovascular hard endpoint). The authors compared the expression of miRNAs and pathway enrichment analysis in cases (n=50) and controls (n=50). Furthermore, the authors examined the association of miRNAs with cardiovascular mortality using multivariate conditional logistic regression analysis. 

As a result, the authors detected 613 miRNAs, among which 5 miRNAs were differentially expressed between cases and controls. Using more stringent cutoffs, only miRNA-129-1-5p was significantly higher in cases than in controls. Pathway enrichment analysis demonstrated enrichment in pathways related with cardiomyocytes and Cardiomyopathy. Finally, higher circulating miR-129-1-5p levels were significantly associated with higher risk of cardiovascular death, independent of potential confounders.

The manuscript is well written. Introduction is quite appropriately written for the background, necessity of the study, and research hypothesis. The methods, results, and discussion are also excellently shown in the manuscript. I have no question about the manuscript.

Reviewer 5 Report

I have one important comment. The finding that miR-129-1-5p is up-regulated in the hemodialysis patients with cardiovascular death can be considered as valuable. The pathway analyses that have been performed regarding the significance of this up-regulation is helpful. One should expect however at least some analyses checking the importance of the pathways popping up in the pathway analyses. Currently, the only "real finding" is the up-regulation of miR-129-1-5p which in my opinion

Round 2

Reviewer 5 Report

The authors answered sufficiently to my commnets